# Lay community mental health workers (cadres) in Indonesian health services: A qualitative exploration of the views of people with mental health problems and their families

Heni Dwi Windarwati[1], Herni Susanti[2]*, Helen Brooks[3], Ice Yulia Wardani[2], Hasniah[4], Mardha Raya[5], Niken Asih Laras Ati[6], Hasmila Sari[7]

1 Department of Mental Health Nursing, Faculty of Health Sciences, Universitas Brawijaya, Malang, Indonesia, 2 Mental Health Nursing Department, Faculty of Nursing Universitas Indonesia, Depok, Indonesia, 3 Mental Health Research Group, Division of Nursing, Midwifery and Social Work, School of Health Sciences, Faculty of Biology, Medicine, and Health, Manchester Academic Health Science Centre, University of Manchester, Manchester, United Kingdom, 4 Ministry of Health Polytechnic Aceh, Aceh Besar, Indonesia, 5 Sambang Lihum Psychiatric Hospital, Banjarmasin, Kalimantan Selatan, Indonesia, 6 Faculty of Nursing, University of Jember, Jember, Indonesia, 7 Mental Health Nursing Department, Faculty of Nursing, Universitas Syiah Kuala, Aceh Besar, Indonesia

* herni-s@ui.ac.id

## Abstract

### Introduction

In community-based mental health services, lay workers recruited and trained to support mental health programs, known as mental health cadres, have an important role in supporting the care of families and people with mental disorders. This study aims to explore the experiences of people with mental disorders and their families about the role of mental health cadres in improving mental health and caring for people with mental disorders.

### Methods

This study employed a qualitative descriptive design for data gathering. Focus groups were conducted between August 2020 and January 2021 with 19 people with mental health difficulties (people diagnosed with schizophrenia) and 25 family members who are the primary caregivers of people with mental disorders from three provinces in Indonesia: West Java, East Java, and Aceh. Participants were purposively sampled with inclusion and exclusion criteria used were people with mental disorders and their families who regularly interact with mental health cadres. Data were analyzed using inductive thematic analysis through six stages of coding and theme development.

### Results

Several themes were identified. The most significant theme was emotional support provided by cadres in terms of reception from cadres about people with mental health problems

**Data Availability Statement:** All data supporting the findings of this study are available within the paper.

**Funding:** This work was supported by the Directorate of Research and Community Engagement University of Indonesia. The funders had no role in study design, data collection and analysis, decision to publish, or preparation of the manuscript.

**Competing interests:** The authors have declared that no competing interests exist.

(59.1%), tangible support in which cadres help people with mental disorders get treatment (52.27%), and cadre roles as duties/mandates (51.36%) was factors that facilitated the success. The finding of this study indicated that cadres were considered to provide a range of different support to people with mental health disorders and their families. In carrying out their role, there were factors that participants felt increased success in implementing the role of cadres. The cadre-patient/family relationship was influenced by perceived shame, trust relationship, and stigma. This research also revealed patient and family expectations about cadres' roles.

## Conclusions

Exploring the experiences of people with mental disorders and their families who received support from cadres could examine the factor that increases success in implementing cadre roles and barriers to mental health services by cadres, which are shame, mistrust, and stigma in the community. Therefore, paying attention to the expectations of people with mental health problems and their families about the cadre's roles in improving mental health services in the community is essential.

## Introduction

Mental health problems are a serious problem and a priority in Indonesia [1]. Severe mental illness (SMI) is a mental, behavioral, or emotional disorder resulting in functional impairment and interfering with major life activities [2]. Recent evidence suggests that as many as 6.7% of households in Indonesia have family members with severe mental illness, where currently, severe mental illness is the second highest cause of disability in people in Indonesia [3, 4]. In addition, about 85% of families who have family members with mental disorders have received treatment at least once to health services, and only 48.9% have received treatment regularly in the last one month [3]. In Indonesia and in many low- and middle-income countries, the burden of health care on patients with mental disorders is compounded by substantially unmet mental health care needs and large disparities in mental health care provision [5, 6].

The family as a caregiver began to assume the vital role of care performed for people with schizophrenia [7]. Family support, acceptance, communication, family hope of recovery for people with schizophrenia, and assistance carried out when seeking treatment, administering patient medication, and helping outreach in the community are the roles of the family during the recovery and rehabilitation process to prevent patient relapse [8, 9]. However, as first-degree relatives, caregivers experience the burden of care during caregiving for patients with schizophrenia receiving treatment in hospital and community settings [10, 11]. A recent study conducted on 136 caregivers showed that caregivers' personal growth was associated with good family functioning and adequate professional support in terms of family psychoeducational intervention positively related to the low burden of care [12, 13]. Effective schizophrenia treatment integrates specific strategies involving a comprehensive treatment plan and mental health professionals [7, 14]. Positive aspects and challenges in caregiving for schizophrenia need adequate support from health professionals and integration into health services.

Integrative collaboration with families and the wider community can improve mental health services in a therapeutic capacity [15]. Community involvement in mental health services is a potential way to overcome the high burden of health care in patients with severe mental disorders [16]. There is a shift from hospital-based mental health services to community-

based mental health services [17]. The shift in health services increases the integration of mental health services and provides opportunities for the community to obtain sustainable mental health services [18, 19]. The potential benefits of this transition are improved access to mental health services, especially for marginalized groups, through connecting and bridging them with health services and strengthening social supports, and improved mental health care outcomes [19–21]. The shift from hospital-based mental health services to community-based health care systems requires elaborate community involvement. The success of community-based mental health care system is influenced by an interdisciplinary mental health team and a responsible person for coordinating services between patients and health systems [18, 22], who is called a lay health worker [23].

Community Health Workers are lay workers recruited from local communities who have received training and supervision from primary care workers. Their roles include promoting health programs to increase the coverage of basic health services in the community [24, 25]. They also facilitate access to care and link the community and the health system [26]. The involvement of Community Health Workers is one way to improve mental health services, reduce treatment gaps, and increase awareness and mental health [27–30]. Patients with mental disorders need lay health workers to help them with treatment management because the failure of treatment plans related to patient medication adherence is caused by often forgetting to take medication or the unavailability of treatment [3]. The study conducted in Ghana showed that community mental health workers (CMHW) could expand mental health services by providing counseling services and home visits as part of their duties [31]. Another study on CMHW in Norway showed that CMHW supports users in the context of life, community participation, and service health systems [32].

In Indonesia, lay mental health workers, called mental health cadres, are representatives of community health centers related to mental health cases in their community. Cadres are expected to have the knowledge and contribute to the treatment and prevention of mental health problems in their community [33]. Mental health cadres who have received mental health training can effectively have the knowledge and ability to implement mental health programs in the community. Mental health cadres play a role in primary prevention through data collection, health education, and motivating clients. In the secondary prevention program, mental health cadres play a role in the early detection and socialization of mental health. Also, in tertiary prevention, cadres, in collaboration with family, remind and monitor patients to take their medication regularly [34–36]. Cadres, as community health workers closest to the community, play a role in assisting patients and their families in controlling the mental health development of patients [37].

Barriers to mental health services include a lack of awareness of professional health services [38]. Patients and families, who are the primary caregivers for people with mental disorders [39, 40], have inadequate knowledge about patients' mental health conditions. They have poor awareness about the availability and access to mental health care and misconceptions about mental health as a problem because of the devil and witchcraft. In addition, as the primary caregiver, the families are at risk of emotional, physical, medical, financial, and social burdens [39]. Families experience heavy burdens and low quality of family life while caring for patients with mental disorders, so they need help to overcome them [41, 42]. These barriers are related to the role of cadres in assisting patients and families in the community, and the perception of patients and families who received assistance from mental health cadres has not been explored.

The role of lay community health workers has been widely studied, but not from the user's point of view. Therefore, this study aimed to explore the experience of people with mental disorders and families regarding the roles of mental health cadres in caring for patients with

mental disorders to deeply examine the barriers to community mental health services and how to increase the role of mental health cadres in implementing and improving mental health services in the community.

## Materials and methods

### Design and context of the study

The study, part of a larger research project titled Exploration of the Role of Mental Health Workers (Cadres) in Community Mental Health Services in Indonesia, explored the experiences of families and people with mental disorders who received support from mental health cadres using a descriptive qualitative and the reporting adhered to the Consolidated Criteria for Reporting Qualitative research (COREQ). This research has been approved by the Ethics Commission of the Faculty of Nursing, the University of Indonesia, with an Ethical Eligibility Letter Number SK-240/UN2.F12.D1.2.1/ETIK 2020. This study followed ethical research principles based on the Belmont Report [43], which includes the principles of beneficence, justice, and respecting human dignity. Written consent was obtained from all participants (patients and families).

### Participants

The study participants were 25 family members and 19 people with mental health disorders. They were recruited from three provinces in Indonesia: West Java, East Java, and Aceh, with details for distribution of focus group discussions in family participants and people with mental disorders presented in Table 1. This study was conducted in these places because Aceh is one of the provinces in Indonesia with a high prevalence of household members with mental health problems. In contrast, the other two provinces have a prevalence of mental health problems below the national average. Therefore, they are pilot projects for implementing the Community Mental Health Nursing Program in Indonesia. Before conducting the study, the researcher asked permission from the local government for each site (West Java, East Java, and Aceh) and coordinated with the Provincial and Regional Health Office. The health office directed us to the community health center for the research location. First, the community health center head identified participants according to predetermined inclusion and exclusion criteria. Then the Head of the community health center, helped by a mental health cadre, made an appointment for a meeting between the researcher participants, people with mental disorders, and families.

Most families who participated in this study were female, from 23 years to 67 years. Almost 90% of the participants had nuclear family relationships with the client, while the others were family relations. The family has cared for the client for around 9.5 years; the longest being 25 years. In addition, the families have interacted with mental health cadres for four years, within

**Table 1. Distribution of focus group discussion family and client.**

| Focus Group Name | Research Sites | Number Or Participatns | Method Of Data |
|---|---|---|---|
| | | Family | |
| FG1 | East Java | 9 | Non Virtual |
| FG2 | Aceh | 9 | Non Virtual |
| FG3 | Bogor | 7 | Non Virtual |
| | | Client | |
| FG4 | East Java | 9 | Non Virtual |
| FG5 | Aceh | 5 | Non Virtual |
| FG6 | Bogor | 5 | Non Virtual |

approximately 3 hours per week. At the same time, all of the people with mental disorders who participated in this study were diagnosed with schizophrenia. Most people with mental disorders were male and average of 36 years. In this study, 63% of people with mental disorders had primary school and did not attend school, while the rest were in junior and senior high school. Almost all participants have been taking the medication regularly in community health centers. Around 68% of clients had a hospitalization history in psychiatric hospitals. Mental health cadres have interacted with clients for nine years and regularly meet clients within 1 hour to 25 hours, with an average of around 3 hours per week. The characteristics of family participants and people with mental disorders in this study are presented in Table 2.

In selecting these participants, a purposive sampling technique was employed to ensure they met the pre-determined criteria and adhered to the research objectives. Criteria for participants in this study were people with mental disorders who lived with their families and did not have a functional impairment (have been able to work independently) [44]. At the same time, the criteria for family members of people with mental disorders were parents, siblings, children, husband/wife, and family relations who are responsible for caring for and meeting the daily needs of people with mental disorders or who play a role in making decisions for them [45]. Participants in this study (people with mental disorders and family members) regularly contacted and interacted with mental health cadres in their respective areas of residence. Inclusion and exclusion criteria for participants were: a) participants are at least 18 years old, as, at this age, a person is considered an adult and can account for the information conveyed during the research process, b) able to communicate well using Indonesian or regional languages (Acehnese/Javanese/Sundanese) which participants and the researchers understand, and c) physically and mentally healthy at the time of the interview. Before the study, family members were assessed for their physical examination and psychological status using the Self Reporting Questionnaire-20 (SRQ-20) [46]. Meanwhile, for people with mental health disorders, a physical assessment and monitoring of symptoms of mental health disorders are carried out, and stated that the symptoms of mental disorders are controlled. In addition, healthcare workers carried out physical and mental health examinations at the community health centers in the participants' respective areas of residence.

The results of the SRQ-20 analysis on families and people with mental health disorders indicated that all participants did not risk experiencing mental and emotional problems (cut-off point <6) [3, 47]. Further analysis of the correlation between SRQ-20 and participant characteristics using the Spearman Rank Test showed that SRQ-20 (risk of emotional and mental health disorders) was significantly related to the estimated time of home visit for cadres in a week (p-value 0.06). In contrast, age (p-value 0.081), gender (p-value 0.138), and length of time cadres interact with family and clients (p-value 0.278) did not significantly associate with the risk of mental-emotional problems.

Therefore, three variables were entered into the Linear Regression analysis to find the variable more related to emotional and mental health problems than the others. After eliminating the length of time families interact with mental health cadres variables, the results revealed that the amount of Adjusted Coefficient of Determination ($R^2$ adjusted) from this model, 13.3% of emotional mental health problems changes were related to the estimated time cadres make home visits in a week (p-value 0.044; B:-0.297) but not statistically significant with variable age (p-value 0.070; B:0.265).

## Data collection

Data for the study were collected from August 2020 to January 2021 using Focus Group Discussions (FGD). Focus group discussion is used to gain an in-depth understanding of the

**Table 2. Participants characteristics.**

| No | Characteristics | Total / Mean | Percentage (%) /(Min-Max) |
|---|---|---|---|
| | **Family** | | |
| 1 | Age (years) | 44.00 | 23–67 |
| 2 | Gender | | |
| | Male | 3 | 12.00 |
| | Female | 22 | 88.00 |
| 3 | Relationship with Client | | |
| | Child | 4 | 16.00 |
| | Parents | 5 | 20.00 |
| | Husband/Wife | 2 | 8.00 |
| | Siblings (brother/sister) | 12 | 48.00 |
| | Cousin/family relations | 2 | 8.00 |
| 4 | Length of time family has cared for the client (years) | 9.56 | 1–25 |
| 5 | Length of time families interact with mental health cadres (years) | 4.12 | 1–18 |
| 6 | Time estimated of cadre home visit time per week (hours) | 3.33 | 0.25–25 |
| 7 | SRQ Score | | |
| | Risk for emotional mental health problems (cut off point $\geq 6$) | 0 | 0 |
| | Did not have emotional mental health problems (cut off point < 6) | 25 | 100 |
| | **Client** | | |
| 1 | Age (years) | 36,47 | 25–54 |
| 2 | Gender | | |
| | Male | 15 | 78.95 |
| | Female | 4 | 21.05 |
| 3 | Education | | |
| | Did not attend school | 5 | 26.32 |
| | Primary School | 7 | 36.84 |
| | Junior High School | 4 | 21.05 |
| | Senior High School | 3 | 15.79 |
| 4 | Medication history at Primary Health Center | | |
| | Regularly | 18 | 94.74 |
| | Not regularly | 1 | 5.26 |
| 5 | Hospitalization history in psychiatric hospitals | | |
| | Have | 13 | 68.42 |
| | Did not Have | 6 | 31.58 |
| 6 | Length of time client interaction with mental health cadres (years) | 9.84 | 1–25 |
| 7 | Time estimated of cadre home visit time per week (hours) | 3.13 | 0.25–25 |
| 8 | SRQ Score | | |
| | Risk for emotional mental health problems (cut off point $\geq 6$) | 0 | 0 |
| | Did not have emotional mental health problems (cut off point < 6) | 19 | 100 |

participants perspectives on the topic in the discussion [48], which was the role of the mental health cadre in mental health services. Focus group discussion is frequently used when participants in the study were selected from a group of individuals purposely and aimed to understand social issues in-depth [49]. This was conducted to portray the views of families and people with mental disorders on mental health cadres who provide in-depth mental health

---

## Box 1. Focus group guideline

1. Opening questions

a What do you think about mental health cadres?

2. Core question

a What do you think about the role of mental health cadres in your area so far?

b What are the things that support and hinder mental health cadres in carrying out their duties?

c What are your expectations about the role of the mental health cadre in the future? What do you need to make these expectations come true?

---

services. The FGD in the family lasted for two hours (in East Java) and one hour (in Aceh and West Java) used Indonesian and local languages (Javanese, Sundanese, and Acehnese) during the discussions. On average, FGD in people with mental health disorders lasted for 60–90 minutes. The Focus group discussions were complete face-to-face at local community venues facilitated by a local moderator with the implementation of a strict COVID-19 health protocol. Moderators and research participants were involved in face-to-face discussions by applying physical distancing, wearing masks, and the discussion was held in an open room with decent air circulation.

Data from the FGD were audio-recorded and field notes to record participants' non-verbal responses and conditions that affect the focus group discussion process was also employed. In addition, a protocol for FGD was used in order to guide the running discussion effectively (Box 1 shows the focus group guideline). The data obtained in the FGD in the form of audio-visual recordings were then transcribed using the Indonesian language.

Before commencing the study, we ensured that the selected participants met the established criteria. The first step of participant selection was screening prospective participants who may participate in this study. Then, healthcare workers responsible for mental health programs at community health centers select participants according to the specified criteria. In this process, we first explained the research objectives and research duration in order the participants are willing to participate voluntarily or withdraw from the study. For ethical reasons, consent forms were signed by the participants. For families, the informed consent was signed before the focus group discussion, while for people with mental disorders, the informed consent was signed accompanied by their family.

### Data analysis

To analyze the data, an inductive thematic analysis was undertaken. Inductive thematic analysis was carried out through six stages of coding and theme development [50]. We did multiple and repeated readings to obtain familiarisation with the data. The first and second authors of this study coded the transcripts independently and then held discussions to determine the final set of codes, in which similar codes would be amalgamated and the duplicated

codes were removed. The final coding framework generated from the discussion process was organized into sub-themes and main themes that aimed to answer research questions. Other authors discussed the potential themes generated at this stage to ensure that the themes fully reflect the data obtained in the research process. The quality criteria for this qualitative research methodology are based on credibility, confirmability, dependability, and transferability [51]. Credibility is equivalent to the internal validity of quantitative research, which shows that research findings are the correct interpretation and have represented the views of research participants [51–53]. In this study, we ensure the trustworthiness of data analysis with peer debriefing through collaboration with all researchers, including non-Indonesian researchers, triangulating data, and providing feedback on data and the resulting themes. All researchers reviewed and agreed on the themes generated in this study. Confirmability indicates neutrality and the extent to which research findings can be confirmed by other researchers [52, 54].

We also described the research results in detail, prepared research notes, and used quotes to present the data to be more transparent. Dependability is consistency and the stability of research findings over time [51, 52, 54], which is improved by going through regular meetings with all researchers to avoid bias, assumptions, and data misinterpretation. We provide complete research procedures so that other researchers can follow the same research process. Transferability refers to the extent to which the results of this study can be transferred to other settings, contexts, and respondents [52, 54], by providing a comprehensive description of the research context (participants and the research process). The research team are academics and mental health professionals, thus enabling an in-depth and comprehensive understanding of mental health problems. The researcher does not have a personal relationship with the participants, so there are no assumptions or personal interests, and the results obtained are purely the opinions of the research participants.

## Results

Six themes were interpreted from the data, namely 1) the type of support provided by the cadre to families and people with mental health disorders, 2) factors that were perceived to facilitate the success of the implementation of the cadre roles, 3) the roles of shame, 4) trust relationships with cadres, 5) stigma, and 6) patient and family expectations of the role of cadres. The findings are also illustrated in Table 3.

### The type of support provided by cadre to families and people with mental health disorders

The key reported roles of cadres identified by participants were monitoring conditions, providing information, and treating mental disorders patients. Participants viewed the cadre as a caring figure who provided care and valued support for people with mental disorders and their families. Participants described cadres in positive roles and considered cadres providing a range of different types of support, including emotional, spiritual, informational, tangible support, social, psychological, and physical support. Emotional support is the action of mental health cadres in giving attention, care, and affection; spiritual support is the effort of cadres to encourage spiritual activity, and informational support is the support given in the form of providing advice and information to improve the health of people with mental disorders. The forms of emotional, spiritual, and informational support provided by the cadres are illustrated in the following participant statements:

**Table 3. Codes, sub-themes, and themes that emerge when exploring the experiences of families and patients in achieving support from cadres.**

| Theme | Sub Theme | Selected code | Frequency of Occurence | Percentage Cover |
|---|---|---|---|---|
| The type of support provided by the cadre to families and people with mental health disorders | Types of Support: Emotional Support | Attention | 22 | 50 |
| | | Give love | 13 | 29.54 |
| | | Care | 18 | 40.91 |
| | | Sympathy with people with mental health disorders | 11 | 25 |
| | | Give support | 5 | 11.36 |
| | | Reception | 26 | 59.1 |
| | Spiritual Support | Encouraging spiritual activity | 8 | 18.18 |
| | | Helping find meaning in life | 11 | 25 |
| | Informational Support | Teaching people with mental disorders to be better | 16 | 36.37 |
| | | Giving advice | 17 | 38.64 |
| | | Cadres inform all mental health activities for people with mental disorders that are carried out at the village hall to families | 7 | 15.91 |
| | Tangible Support | Help get treatment for people with mental health disorders | 23 | 52.27 |
| | | Supervise / control the accuracy of taking medication | 21 | 47.73 |
| | | Helps quickly recognize signs of relapse | 7 | 15.91 |
| | | Facilitate when people with mental disorders need to be referred | 6 | 13.64 |
| | | Providing services to the community | 4 | 9.09 |
| | Occupational Support | Teaching skills | 6 | 13.64 |
| | | Helping people with mental disorders Decent work | 8 | 18.18 |
| | | Teaching art | 7 | 15.91 |
| | Sosial Support | Invite socialization | 8 | 18.18 |
| | Psychological Support | Family is happy | 10 | 22.73 |
| | | Family feeling calm | 3 | 6.82 |
| | Physical Support | Feel physically fit | 9 | 20.45 |
| | | The family feels there is a positive activity | 4 | 9.09 |
| | Support Time Provided by Cadres | Support time | 13 | 29.54 |
| | | Support duration | 3 | 6.82 |
| Factors that were perceived to facilitate the success of the implementation of the cadre roles | Internal factors: Positive Attitude of Mental Health Cadres | Friendly | 5 | 11.36 |
| | | Attention | 14 | 31.82 |
| | | Full of love | 8 | 18.18 |
| | Commitment | Duties/mandate | 16 | 51.36 |
| | | Responsible | 10 | 22.73 |
| | External Factors: Cadre's Close Relationship with Patients and Families | Family | 7 | 15.91 |
| | | Relation | 2 | 4.54 |
| | Community Leader Support | Feeling community leaders are responsible and monitor activity | 4 | 9.09 |
| | Availability of Facilities and Infrastructure for Activities | Integrated Healthcare Center | 5 | 11.36 |
| | | Village meeting hall | 2 | 4.54 |
| The roles of shame | Perceived shame | Shy | 3 | 6.82 |
| | | Feel ashamed | 4 | 9.09 |
| | Inferior | Don't want meet others | 4 | 9.09 |

*(Continued)*

**Table 3.** (Continued)

| Theme | Sub Theme | Selected code | Frequency of Occurence | Percentage Cover |
|---|---|---|---|---|
| **Trust relationships with cadres** | Rejection | Don't want to meet cadres | 2 | 4.54 |
| | | Patient sleeping | 2 | 4.54 |
| | | Don't want to take medicine, feel no pain | 6 | 13.64 |
| | Family Mistrust | Family refuses | 1 | 2.27 |
| | | Closed Family | 3 | 6.82 |
| **Stigma** | Patients in Crisis | Feeling incapable | 4 | 9.09 |
| | Conflict | Offended family | 1 | 2.27 |
| | Stigma | Labelled negative | 4 | 9.09 |
| | | Considered useless | 2 | 4.54 |
| **Patient and Family Expectations on Cadre Roles** | Mental Health Cadre's Ability | Commitment in providing service | 9 | 20.45 |
| | | Communication skills | 5 | 11.36 |
| | | Sustainability of activities | 6 | 13.64 |
| | | Partnership | 6 | 13.64 |
| | Forms of Cadre Support to Families and Patients | Reduce stigma | 5 | 11.36 |
| | | Provide information | 12 | 27.27 |
| | | Maintenance assistance | 10 | 22.73 |
| | | Financial support | 6 | 13.64 |
| | | Work skills | 8 | 18.18 |
| | | Home visit | 9 | 20.45 |
| | | Spiritual support | 3 | 6.82 |
| | Mental Health Cadre's Attitude | Attention | 7 | 15.91 |
| | | Empathy | 6 | 13.64 |
| | | Communicative | 5 | 11.36 |
| | Training Support for Cadre | Training for cadre | 2 | 4.54 |
| | | Advanced training for cadres and families | 2 | 4.54 |

*"The cadres care about my family, always regularly doing home visits. They feel sorry for me because many of my family members are sick. . .So they (cadres) are also there to comfort me so that I do not feel sad and my burden is less"*

*(Family B3—female, 41 years)*

*"The cadre always looked after me and my family continually monitored whether the medicine was taken regularly. They also like to provide advice so that the patient gets better quickly"*

*(Family B3- female, 41 years)*

*"Cadre who guides us, tells us to recite the Koran, and tells us to pray. . ."*

*(Patient B1—male, 40 years)*

Tangible support is practical and visible support provided by cadres to families and people with mental health problems. Cadres help patients with mental disorders get treatment, control the accuracy of taking medication, help families recognize signs of patient relapse, facilitate when patients need to be referred, and provide services to the community. It is depicted in the following vignette:

*"Cadres usually help us remind whether my family (patient) have been taking medicine. If I didn't grab medicine, she would grab medicine for my family, so I do not have to go to Community health centres"*

*(Family A2—female, 31 years)*

*"Cadre regularly visits the house, or if I report signs of recurrence, she (cadre) immediately responds"*

*(Family A3—female, 49 years)*

Mental health cadres also encourage people with mental disorders to have jobs, promote and provide social support. Cadres teach skills, teach art, and help people with mental disorders who are independent and fit to work to have jobs. Cadres taught people with mental disorders skills for making bags, doormats, flower arrangements, making cake boxes, and other skills such as gymnastics. It is shared by Patient A5 in the discussion:

*"Teach skills. . . making. . .bags, making doormats"*

*(Patient A5—male, 35 years)*

Families and people with mental disorders also received psychological support and physical support from mental health cadres, which improves family peace and health. Psychological support improves psychological well-being, while physical support includes the facilitation of cadres to support the implementation of people with mental disorders activities.

*"I feel happy. There are activities and sports which are carried out at the village hall regularly"*

*(Patient A5—male, 35 years)*

Mental health cadres regularly interacted with participants every day, one to two times a week, and at least once a month to support families and people with mental disorders. In addition, participants and cadres frequently have houses adjacent so the cadres can make more visits. The excerpts below illustrate such conditions:

*"The cadres service is good. Held at the village hall, every two weeks."*

*(Family A1—female, 51 years)*

*"Almost every day, she (cadres) comes. After all, our houses are close"*

*(Family B8—female, 29 years)*

## Factors that were perceived to facilitate the success of the implementation of the cadre roles

The success of cadres in carrying out their duties was perceived by participants to be mediated by a range of factors (internal and external factors). The positive attitude of mental health cadres and commitment of mental health cadres are internal factors perceived by participants that support the success of cadres in carrying out their duties. Mental health cadres show a friendly, caring, and loving attitude to patients and families when making visits and carrying out their duties, thereby increasing family acceptance of the role of cadres. These two vignettes portray such situations:

*"cadres are friendly people, so the family accepts them"*

*(Patient B1—male, 40 years)*

*"she feels he has a responsibility to our family"*

*(Family B5—female, 23 years)*

In addition, participants described cadre commitment as dedication that causes cadres to do their responsibility. It showed by mental health cadres in the form of obligation and responsibility in providing services is also a major supporter of the successful implementation of the duties of cadres in the community. In carrying out their duties, cadres make home visits from one patient's house to another patient's house, besides that, mental health cadres also visit the patient's family. Patient B1, B3, and Family B4 contend that:

*"She (cadres) regularly visit and monitor the condition of my family. I feel that she are responsible for my family, maybe because it is a mandate and duty as a cadre"*

*(Family B4—female, 36 years)*

*"Cadres are always there when we need them"*

*(Patient B3—male, 33 years)*

*"Cadre came for guided us to get well soon"*

*(Patient B1—male, 40 years)*

Participants felt close to cadres because cadres were from their communities and viewed cadres have their own experiences with mental health problems. The close relationship is an external factor that supports cadres in carrying out their duties properly, for example, mental health cadres who are still family members or closest relatives. Cadres who are relatives (cousins) or family (wife/husband/close family) of people with mental disorders have better relationships because they are more open with cadres and usually have more intensive interactions because they live closer to participants. Patient B1, A4, and Family B9 shared that:

*"My wife is a cadre, and I have become calmer and easier to tell something"*

*(Patient B1—male, 40 years)*

*"Cadres are my family; I know her; her house was close to me"*

*(Patient A4—male, 28 years)*

*"In my opinion, cadres who are still family members are easier, usually easier to close and easier to open up"*

*(Family B9—female, 32 years)*

The support of community leaders, as well as the availability of facilities and infrastructure for the implementation of cadre activities are other external factors that play a role in helping cadres carry out their duties. Patient A3 and Family A5 stated that:

*"There are routine cadre activities at the village hall"*

*(Patient A1—male, 38 years)*

"*I see the village head always monitors our activities (cadre and patients/family)*"

*(Family A5—female, 62 years)*

## The roles of shame

While receiving support from mental health cadres, several participants mentioned that they felt ashamed of the mental health problems they or their families were experiencing. The perceived shame felt by the patient that caused the patient to feel inferior also made it difficult for the cadres to carry out their duties because the patient did not want to meet the cadres. It is depicted that:

"I feel shame to meet others. I don't want to meet them."

*(Patient B5—male, 40 years)*

"I feel ashamed to be visited by cadres because if cadres visit, it means that there was a mental disorder at my home"

*(Familyt B5—female, 23 years)*

## Trust relationships with cadres

In carrying out their role, participants perceived several obstacles encountered by the cadres in implementing their duties. Barriers to cadres in carrying out their duties are influenced by two factors, namely factors from patients and factors from families. When carrying out their duties, mental health cadres sometimes experience rejection from patients, such as excluding from meeting with cadres, not wanting to take medicine, or patients not feeling sick, making it difficult for cadres to carry out their duties properly. Another obstacle that comes from the patient's family is the conflict between the family and the family, who does not trust the mental health cadres. Sometimes, the patient and family do not trust the mental health cadres. This trust relationship with cadres brings up barriers to cadres carrying out their duties. This trust relationship easily improved when cadres were family or relatives of the patients. In the FGD, Family B7 shared that:

*"My family didn't want to meet the cadres and didn't want to take medicine because they didn't feel sick, so when the cadres came, they refused."*

*(Family B7—female, 29 years)*

*"Cadres are close to us, my family, so I can be more open and more trusting"*

*(Family A2—female, 31 years)*

## Stigma

Participants felt that the support of cadres for patients and families with mental health problems was related to stigma in the community. Patients' families sometimes feel offended and labelled negative by community when cadres make home visits, so the patient's family is not open and does not accept advice from cadres. The existence of stigma from the community to the patient and family causes the patient and their family to feel ashamed and worsens the services provided by the cadres. This is because the family feels reluctant to be visited by mental health cadres. Family B5, B7, and B9 shared in the FGD that:

*"Embarrassed because there is a stigma in the community if a cadre is visited, it means that someone is crazy at home."*

*(Family B5—female, 23 years)*

*"If a cadre comes to the house, I feel offended"*

*(Family B7—female, 29 years)*

*"I felt offended when the cadres visited, so I just said that if there was no family who was sick and had recovered"*

*(Family B9—female, 32 years)*

### Patient and family expectations on cadre roles

Patients and families hope that mental health cadres can support families in caring for patients with mental disorders. The ability of mental health cadres is demonstrated by the commitment of cadres to providing care, sustainability of activities, partnerships with health centers and villages and the skills of mental health cadres in communicating with families and patients. With good communication skills, families and patients hope that communication with cadres can run smoothly and information about mental health can be conveyed properly. In the discussion, Patient B4, Patient C1 and Family C1 voiced that:

*"I hope that our activities (with cadres) will run more smoothly."*

*(Patient B4—male, 27 years)*

*"Communication needs to be improved because sometimes things are correct, and things are not. So hopefully, we can discuss health."*

*(Patient C1—male, 33 years)*

*"More for communication with the health center and village"*

*(Family C1—female, 62 years)*

Another expectation by families and patients towards mental health cadres is the existence of other forms of support provided by mental health cadres. The form of support in question is the provision of information, care assistance, financial support and work skills, home visits, spiritual support, and hopes to reduce the stigma on families and patients with mental disorders. Although several forms of support have been provided by mental health cadres, for example, in the form of providing information, care assistance, paper skills support, home visits, and spiritual support, patients and families hope that this form of support can be improved for the better with a slightly longer duration. Families perceived that when cadres have more contact with patients, cadres can see the development of their families (people with mental disorders), monitor patient independence, and assess patient signs and symptoms. Families also perceived that when cadres often visit patients, they can improve patient recovery. Families and patients also hope that apart from being given advice, mental health cadres can provide knowledge related to mental health. It is shared that:

*"Cadres can visit so they know how far the progress of our children"*

*(Family A4—male, 67 years)*

*"Even though the patient is independent, Cadres still has to visit"*

*(Family B6—female, 38 years)*

*"When visiting the patient's family, it is hoped that the patient will also be asked their signs and symptoms."*

*(Family B9—female, 32 years)*

"*Often see my son at my house to increase his improvement*

*(Family C7—female, 25 years)*

*"Family expectations for Cadre. . . Providing advice and knowledge."*

*(Family B6—female, 38 years)*

Stigma causes cadres to experience obstacles in providing services. Families and patients hope that cadres can provide understanding to families and communities so that families are more open and do not feel excluded. Families hope cadres can communicate about stigma and provide counseling to families to reduce stigma in society. Family B2 shared that:

*"Often provides counseling to families. It means working with the village head to be able to provide counseling to the community regarding the stigma of mental disorders. . .so that stigma is lost in the community related to mental disorders"*

*(Family B2—female, 38 years)*

Mental health cadres are expected to have a caring attitude towards their families and patients and have a sense of empathy. The family hopes that the cadres can have more attention and a better attitude to patients. In addition, effective communication between cadres and families and patients' needs to be further improved. Family C3 and Family C1 contend that:

*"Pay more attention to patients so that they are even better"*

(Family C3—*female, 47 years*)

"*More to communication with cadres*"

*(Family C1—female, 62 years)*

Families also hope that mental health cadres will receive support from the health system in the form of training for mental health cadres. With the training of mental health cadres, families hope that cadres can carry out their roles better and can provide further training for families who care for patients with mental disorders at home.

*"So that the mothers in this Community Health Center give more frequent training to Cadres so that they are better in the community"*

*(Family B6—female, 38 years)*

## Discussion

### Cadre support to families and people with mental health disorders

The forms of support provided by cadres to families include emotional support, spiritual support, and informational support. Cadres can approach families with mental patients because cadres are in an environment with the same cultural background, using open and friendly communication strategies [55]. Cadres help families and people with mental disorders by providing information, referring to Community Health Center, controlling treatment, and other social assistance [37]. Improving knowledge and insight about schizophrenia through psychoeducation promotes social functioning, reduces relapse, and encourages medication treatment adherence in people with schizophrenia [56]. In addition, the care of cadres as social support can improve the quality of life of patients with mental disorders [57]. Families feel heard and assisted by cadres in providing care to family members with mental disorders [58]. Families find it helpful when cadres teach patients to worship and train families to accompany them [55]. Cadres also provide skills training to patients and then distribute the patient's work with entrepreneurs or non-governmental organizations [59]. These various efforts are a form of cadre support so that families and patients with people with mental disorders feel helped. Families and patients benefit from having mental health cadres in their area of residence.

### Supporting factors for cadre success in carrying out tasks

Supporting factors consist of internal factors (friendly, caring, loving attitude to patients and families) and external factors (closeness of cadres to patients and families, and support from community leaders). Cadres play a role in connecting health workers with patients and families to help the successful resolution of mental health problems in the community [60]. Cadres show a friendly and empathetic attitude to patients and families when interacting, which affects the success of the approach [55]. In addition, cadres' attitude of attention and affection to patients makes families more cooperative in receiving the information conveyed [61]. The motivation and sincerity of cadres in doing service can be seen from the efforts to carry out treatment and home visits so as to increase the success of the community mental health program [62]. Health workers provide support in the form of assistance to cadres every time they carry out tasks such as counseling and treatment for patients with severe mental disorders [63]. The task of mental health cadres in the community is also assisted by the collaborative role of community leaders and across sectors to be more effective [64]. Motivation and sincerity from within cadres accompanied by support from other people, both health workers, community leaders, and cross-sectors, affect the success of solving community mental health problems.

### Factors inhibiting the implementation of cadre duties

Mental health cadres experience rejection from patients, such as patients not wanting to meet mental health cadres, feeling embarrassed, not wanting to take medicine, not feeling sick, and family conflicts that make it difficult for cadres to carry out their duties. Patients sometimes refuse the presence of mental health cadres because they feel that cadres are not health workers who can provide assistance [65]. Stigma still high in society towards mental disorders makes families and patients hide their pain and not feel sick [66]. The stigma of mental illness in the community and health services hampers the quality of mental health services [67]. Conflicts in the family, such as relationships between family members or work problems, can have a negative impact on mental health and health-seeking behavior [68]. Caring for family members with mental health disorders is challenging for caregivers and causes varying degrees of care

burden requiring routine psychological family intervention [10, 13]. For example, the caregiver burden was higher in community settings (28.28%) and in caregivers of individuals with psychosis (35.88%) [11]. Another inhibiting factor is the attitude of patients who refuse treatment. The patient refuses to take medication because he has already undergone treatment and has not recovered or does not feel sick, so other collaborative efforts are needed in approaching the patient and family [69]. The thing that hinders the success of the cadre's duties is the patient and the family in responding to the treatment process for mental disorders.

### Patients and family expectations on cadre roles

Patients and families hope that communication with mental health cadres can run smoothly and information about mental health can be conveyed properly. Patients and families also hope to receive support or attention from cadres, reduce community stigma, and cadres receive training. Training could grow cadres' confidence and their relationships with others; this can support the effective treatment of schizophrenia, which requires integrating psychosocial intervention strategies and optimal medication use [14, 70]. The role of cadres in conducting counseling, assisting in monitoring, and implementing therapy is very beneficial for patients and families so that families hope that the relationship between cadres and patients is maintained [71]. Adequate professional support from cadres is positively related to good family functioning, so it has a positive impact on reducing subjective and objective burdens of care [12]. Also, families need social support from cadres as people who are closer and help communicate with professional health workers [72]. Family involvement and social support, such as cadres in treating mental disorders, influence the patient's confidence to recover [73]. The main hope felt by patients and their families is the reduction of stigma in society because stigma affects the family's attitude in receiving a diagnosis and also efforts to seek health care for patients with mental disorders [66]. Patients and families hope to continue to receive positive support from cadres and health workers, reducing societal stigma so that patients can be independent and productive.

### Conclusions

Families and patients with mental disorders feel the cadres' commitment and good attitude in helping families monitor conditions, provide information, and treat patients with mental disorders. Data from patients and families show expectations for cadres to improve their ability to communicate and deliver care, commitments related to the continuity of activities, and partnerships with Community Health Center and cadres inhibiting factors in carrying out their duties. First, in the form of refusal, the patient does not want to take medicine and violent behavior. The family factor refuses to be visited because they feel ashamed due to the stigma related to mental disorders from the community. This study contributes to achieving the Global Goals of the 2030 Agenda for Sustainable Development, which was ensuring healthy lives and promoting well-being for all at all ages through the role of mental health cadres in supporting the care of patients with mental disorders and their families, thereby increasing family well-being and bridging mental health services for people with mental disorders. To optimize the role of cadres in society, a more structured training program is needed to improve the capabilities of cadres. In addition, programs that target the improvement of community literacy related to mental disorders are needed.

### Acknowledgments

The authors would like to express their gratitude to all who contributed to this research.

## Author Contributions

**Conceptualization:** Heni Dwi Windarwati, Herni Susanti.

**Data curation:** Ice Yulia Wardani,  Hasniah, Mardha Raya.

**Formal analysis:** Ice Yulia Wardani,  Hasniah, Mardha Raya.

**Methodology:** Heni Dwi Windarwati, Herni Susanti.

**Validation:** Helen Brooks.

**Writing – original draft:** Niken Asih Laras Ati, Hasmila Sari.

**Writing – review & editing:** Heni Dwi Windarwati, Herni Susanti, Helen Brooks, Niken Asih Laras Ati, Hasmila Sari.

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
