## [Decision Letter · Decision Letter 0]

22 Mar 2023

PONE-D-22-34567Lay community mental health workers (cadres) in Indonesian health services: A qualitative exploration of the views of people with mental health problems and their familiesPLOS ONE

Dear Dr. Susanti,

Thank you for submitting your manuscript to PLOS ONE. After careful consideration, we feel that it has merit but does not fully meet PLOS ONE’s publication criteria as it currently stands. Therefore, we invite you to submit a revised version of the manuscript that addresses the points raised during the review process.

Please note that we have only been able to secure a single reviewer to assess your manuscript. We are issuing a decision on your manuscript at this point to prevent further delays in the evaluation of your manuscript. Please be aware that the editor who handles your revised manuscript might find it necessary to invite additional reviewers to assess this work once the revised manuscript is submitted. However, we will aim to proceed on the basis of this single review if possible. 

The reviewer’s comment may be seen below. In particular, the reviewer has provided many useful comments to further improve the reporting and discussion of the manuscript.

Could you please revise the manuscript to carefully address the concerns raised?

We look forward to receiving your revised manuscript.

Kind regards,

Lucinda Shen, MSc

Staff Editor

PLOS ONE

Journal Requirements:

Reviewers' comments:

Reviewer's Responses to Questions

**Comments to the Author**

1. Is the manuscript technically sound, and do the data support the conclusions?

Reviewer #1: Partly

2. Has the statistical analysis been performed appropriately and rigorously? 

Reviewer #1: No

3. Have the authors made all data underlying the findings in their manuscript fully available?

Reviewer #1: No

4. Is the manuscript presented in an intelligible fashion and written in standard English?

Reviewer #1: Yes

5. Review Comments to the Author

Reviewer #1: The current qualitative study conducted on a small sample aimed to explore the experiences of people with mental disorders and their families about the role of mental health workers in improving users’ mental health and in providing support to their caregivers. Given the relevance of roles assumed by families in the context of community psychiatric care in looking after relatives affected by mental disorders, the paper provides scientific contribution to the panorama of studies investigating on the role of family members in caring for relatives affected by mental disorders.

Globally, the clarity of the manuscript is overall good enough in the language style, syntax, and sentences construction. The topic is interesting enough and qualitatively good to be published. However, some changes and explanations to improve quality of manuscript for his acceptance should be done.

“Abstract section”

In the description of the methods, the authors should also report the site or sites of recruitment and the mental disorders that predominantly characterized the sample.

In the results section, the authors could report the most significant data, in quantitative terms (relevant statistical data).

The keyword "caregiver or “caregivers " could be entered.

Introduction section

In the ‘introduction section’ the bibliography is appropriate and updated.

However, it would be useful to mention, for example, some studies that have focused on the role of 1) the family as a resource for the rehabilitation process of the family member suffering from mental disorder, and of 2) the scientific evidence based interventions.

There are several evidences of the effectiveness of psychoeducational interventions

involvement of family members aimed at mitigating the effects of the disease, in particular, schizophrenia (1,2,3,4,5,6 listed below). The authors should integrate the bibliography reported, considering the development of EBM psychoeducational interventions aimed at family members of people with serious and disabling mental illnesses.

For example, a very recent study conducted on 136 caregivers showed that caregivers’ personal growth was associated with good family functioning, but also, adequate professional support. suggesting as the many challenges and positive aspects associated with caregiving they should be recognized by mental health services and integrated into routines clinical assessment and intervention framework.

Row 74. In the sentence “…….arrangements with families and the wider community in a therapeutic capacity..” it is unclear what the authors mean by ‘arrangements’.

“Materials and methods section”

Row 135: The authors contextualize this study in a "larger research project"; the authors could clarify better and report more information on the mentioned project.

Row 148: The authors specify the countries in which the recruitments were carried out, but neither the structure in which the patients were recruited nor the number of accesses and the percentage of patients recruited on the basis of them is specified.

Rows 149-150. The authors report “…… with details for 149 family participants and people with mental disorders presented in table 1.” Socio-demographic and clinical data of the sample is not present in the manuscript. Furthermore, Table 1 reports “Distribution of Focus Group Discussion Family and Client”. Could the authors clarify this inconsistency? It would be appropriate to report the clinical data of the sample (patients and family members), the diagnoses, as well as the role of the caregivers (i.e. mother, father, husband, wife, brother, sister, etc.) either in the table or in the text.

Regarding Table 2 , it could be implemented with the percentages of the identified themes.

Row 184: The authors reported that they had administered the Self Reporting Questionnaire-29 (SRQ-29) scale to family members without any description; it would be useful to know which constructs/dimensions the questionnaire evaluates, related psychometric information and examples of items. In addition, it could be interesting to carry out regression analyses with the socio-personal data collected and questionnaire dimensions.

‘Discussion and Conclusions section’

In the discussion, as well as in the introduction, the authors should integrate the discussion and comment on the results by citing family psychoeducational approaches ( 1,2,3,4,5,6 listed below).

In the ‘Conclusions section’ the Authors should report how their study contributes to achieving the Global Goals of the 2030 Agenda for Sustainable Development.

References

References are reported inconsistently. For example, some contain the 'DOI' while others do not; some (i.e. citation 39) report volume and page numbers, others do not. Greater homogeneity and coherence between references would be needed

1. https://pubmed.ncbi.nlm.nih.gov/36713911/

2. https://pubmed.ncbi.nlm.nih.gov/29942416/

3. https://pubmed.ncbi.nlm.nih.gov/36553947/

4. https://pubmed.ncbi.nlm.nih.gov/24879572/

5. https://pubmed.ncbi.nlm.nih.gov/9565182/

6. https://pubmed.ncbi.nlm.nih.gov/21147896/

6. PLOS authors have the option to publish the peer review history of their article (what does this mean?). If published, this will include your full peer review and any attached files.

Reviewer #1: No

---

## [Author Response · Author response to Decision Letter 0]

13 Jun 2023

Thank you for your email dated March 24, 2023, and reviewer comments for the manuscript titled “Lay community mental health workers (cadres) in Indonesian health services: A qualitative exploration of the views of people with mental health problems and their families” (Manuscript ID: PONE-D-22-34567). As directed by your correspondence, we are submitting a substantially revised manuscript with three columns of tables that respond to each point raised by the academic editor and reviewer. We also submit a marked-up copy of our manuscript highlighting changes made to the original (track changes version) and an unmarked version of our revised paper without tracked changes. So again, we thank you for the opportunity to submit this revised manuscript. We look forward to your feedback.

Response to the reviewer comments

Abstract

1. We thank the reviewers for taking the time to assess our manuscript.

We are grateful for the appreciation given by the reviewers for our manuscript.

We have ensured to improve the manuscript based on suggestions and input from reviewers.

2. We have added predominantly characterized mental disorders and sites of recruitment in the abstract.

In this study, the mental health problems experienced by participants were schizophrenia (100%/19 participants). Nineteen people with mental health difficulties are people diagnosed with schizophrenia.

The recruitment sites in this study are three provinces in Indonesia: West Java, East Java, and Aceh.

3. We have added the most significant theme in the quantitative term.

4. We have inserted the new keyword "caregiver."

Introduction section

1. We have added a paragraph explaining the reviewer's suggestions regarding the family as a resource for the rehabilitation process of people with mental disorders, evidence of the effectiveness of psychoeducational interventions aimed at family members, and suggested challenges and positive aspects associated with caregiving by integrated mental health services.

2. What we mean is a collaboration with families and communities.

We have rearranged the sentence to

"Integrative collaboration with families and the wider community can improve mental health services in a therapeutic capacity."

Materials and methods section

1. We have inserted the title of the project in the manuscript. 

2. The participant selection process was carried out based on the following process:

We asked permission from the local government for each site (West Java, East Java, and Aceh) and coordinated with the Provincial and Regional Health Office. The health office directed us to the community health center for the research location. The Head of the community health center identified participants according to predetermined inclusion and exclusion criteria. Then the Head of the community health center, helped by a mental health cadre, made an appointment for a meeting between the participants and the researcher.

3. We have added data about the characteristics of families and people with mental health problems who participated in this study as well as diagnoses of people with mental disorders (Table 2).

4. We have added the percentages of identified themes in Table 3

5. We identified the SRQ in the participants of this study using the SRQ-20. We apologize for the mistakes in writing our manuscript.

We performed an SRQ analysis and showed that all participants (25 families and 19 people with mental disorders) had scores below the cut-off point (<6).

We also performed correlation analysis using the Spearman Rank and showed that only the variable estimated time of home visit for cadres in a week had a significant relationship with SRQ (p-value 0.06). In contrast, age (p-value 0.081), gender (0.138), and length of time cadres interact with family and clients (0.278) did not significantly associate with the risk of mental-emotional problems.

We conducted regression analysis and obtained the model. After eliminating the length of time families interact with mental health cadres variables, the results revealed that the amount of Adjusted Coefficient of Determination (R2 adjusted) from this model, 13.3% of emotional mental health problems changes were related to the estimated time cadres make home visits in a week (p-value 0.044; B:-0.297) but not statistically significant with variable age (p-value 0.070; B:0.265).

Discussion and Conclusions section

1. We have integrated the discussion section by citing literature following recommendations from reviewers.

2. We have inserted how our study contributed to SDGs 3: 

This study contributes to achieving the Global Goals of the 2030 Agenda for Sustainable Development which was ensuring healthy lives and promoting well-being for all at all ages through the role of mental health cadres in supporting the care of patients with mental disorders and their families, thereby increasing family well-being and bridging mental health services for people with mental disorders.

References:

We have revised the inconsistently and made sure that references have homogeneity

---

## [Editor Report · Decision Letter 1]

31 Jul 2023

Lay community mental health workers (cadres) in Indonesian health services: A qualitative exploration of the views of people with mental health problems and their families

PONE-D-22-34567R1

Dear Dr. Heni Dwi Windarwati

We’re pleased to inform you that your manuscript has been judged scientifically suitable for publication and will be formally accepted for publication once it meets all outstanding technical requirements.

Kind regards,

Kishor Adhikari, Ph.D.

Academic Editor

PLOS ONE

Additional Editor Comments (optional):

Thank you for addressing all the concerns and querries. Now, the revised version seems okay for me.